# Comparative Metabolomic Analysis of *Moringa oleifera* Leaves of Different Geographical Origins and Their Antioxidant Effects on C2C12 Myotubes

**DOI:** 10.3390/ijms25158109

**Published:** 2024-07-25

**Authors:** Roberta Ceci, Mariateresa Maldini, Piergiorgio La Rosa, Paolo Sgrò, Garima Sharma, Ivan Dimauro, Mark E. Olson, Guglielmo Duranti

**Affiliations:** 1Laboratory of Biochemistry and Molecular Biology, Department of Movement, Human and Health Sciences, Università degli Studi di Roma “Foro Italico”, Piazza Lauro De Bosis 6, 00135 Roma, Italy; roberta.ceci@uniroma4.it; 2SCIEX Italia S.r.l., Via Montenapoleone, 8, 20121 Milano, Italy; mariateresa.maldini@sciex.com; 3Division of Neuroscience, Department of Psychology, Sapienza University, Via dei Marsi 78, 00185 Roma, Italy; piergiorgio.larosa@uniroma1.it; 4Laboratory of Endocrinology, Department of Movement, Human and Health Sciences, Università degli Studi di Roma “Foro Italico”, Piazza Lauro De Bosis 6, 00135 Roma, Italy; paolo.sgro@uniroma4.it; 5Department of Botany, University of Delhi, Delhi 110007, India; garimasharma1707@gmail.com; 6Laboratory of Biology and Human Genetics, Department of Movement, Human and Health Sciences, Università degli Studi di Roma “Foro Italico”, Piazza Lauro De Bosis 6, 00135 Roma, Italy; ivan.dimauro@uniroma4.it; 7Instituto de Biología, Universidad Nacional Autónoma de México, Tercer Circuito de CU S/N, Ciudad de México 04510, Mexico; molson@ib.unam.mx

**Keywords:** *Moringa oleifera* leaf extract (MOLE), LC-MSMS ZenoTOF 7600 system metabolomics analysis, C2C12 skeletal muscle cells, oxidative stress

## Abstract

*Moringa oleifera* is widely grown throughout the tropics and increasingly used for its therapeutic and nutraceutical properties. These properties are attributed to potent antioxidant and metabolism regulators, including glucosinolates/isothiocyanates as well as flavonoids, polyphenols, and phenolic acids. Research to date largely consists of geographically limited studies that only examine material available locally. These practices make it unclear as to whether moringa samples from one area are superior to another, which would require identifying superior variants and distributing them globally. Alternatively, the finding that globally cultivated moringa material is essentially functionally equivalent means that users can easily sample material available locally. We brought together accessions of *Moringa oleifera* from four continents and nine countries and grew them together in a common garden. We performed a metabolomic analysis of leaf extracts (MOLE) using an LC-MSMS ZenoTOF 7600 mass spectrometry system. The antioxidant capacity of leaf samples evaluated using the Total Antioxidant Capacity assay did not show any significant difference between extracts. MOLE samples were then tested for their antioxidant activity on C2C12 myotubes challenged with an oxidative insult. Hydrogen peroxide (H_2_O_2_) was added to the myotubes after pretreatment with different extracts. H_2_O_2_ exposure caused an increase in cell death that was diminished in all samples pretreated with moringa extracts. Our results show that *Moringa oleifera* leaf extract is effective in reducing the damaging effect of H_2_O_2_ in C2C12 myotubes irrespective of geographical origin. These results are encouraging because they suggest that the use of moringa for its therapeutic benefits can proceed without the need for the lengthy and complex global exchange of materials between regions.

## 1. Introduction

The natural products of *Moringa oleifera* Lam. (*M. oleifera*, family Moringaceae, order Brassicales, Figure 1) have been the subject of significant research recently, and their biological effects have been extensively investigated in various in vivo and in vitro models [1,2,3,4,5]. Many bioactive molecules are present in different parts of the plant, such as the seeds, roots, fruits, and leaves. These molecules demonstrate many beneficial features, acting as antioxidants, antimicrobial agents, and metabolic regulators [6,7,8]. Samples from *Moringa oleifera* have antioxidant features that are due to the presence of isothiocyanates, tannins, saponins, flavonoids, and terpenoids, especially in the leaves [6,7,8,9]. Given the promise of the extraordinary efficacy of its bioactive molecules, in particular some peculiar isothiocyanates, *M. oleifera* is of interest in multiple applications worldwide. The isothiocyanate moringin, produced following hydrolysis of the glucosinolate glucomoringin by the enzyme myrosinase, provides potent anti-inflammatory and indirect cytoprotective antioxidant activity. This is of particular interest in the use of *Moringa oleifera* extracts for therapeutic purposes for various pathologies [9,10,11].

In the global implementation of *M. oleifera* in dietary and other applications, a persistent question regards the geographical variation in applied properties. The oldest documentation of *M. oleifera* comes from ancient Indian texts mentioning its use in medicine [12]. While wild populations that are unequivocally ancestral to today’s cultivated *M. oleifera* have yet to be located [10], documented millennia-old use makes it almost certain that *M. oleifera* had its original wild range in what is now India. Because *M. oleifera* is well established in traditional cuisine in the Philippines, and because one of the most common names in the Philippines, malunggay, is very similar to “murungai,” a name that has been documented in use in Tamil Nadu for hundreds of years, it seems likely that *M. oleifera* was moved to the Philippines at least hundreds of years ago. More recent movements in the colonial period saw moringa moved to the New World tropics, across the Pacific, and into Africa, via Indian railroad workers in English African colonies. Overlain by this pattern of distribution of *M. oleifera* resulting from human migration are more recent movements. These more recent movements have intensified in the last 20 years as agriculturists purchase seeds from international sources and import them to their home countries to grow moringa for commercial use. Given this history of movement of plant material, it is not clear to what degree *M. oleifera* grown in different parts of the world has different applied properties.

Across *M. oleifera’s* vast cultivated range across the world’s tropics, there are two different possible scenarios regarding variation in its applied properties. These two scenarios have very different implications for scientific research on the applied use of *M. oleifera*.

One situation considers the genetic differentiation that occurred in the expansion of the species into the tropical zone. Natural selection under the particular conditions of soil, climate, and even agricultural or management practices, could lead to local adaptation, given that *M. oleifera* is grown so extensively in such geographically distant locales with such differing conditions. Local adaptation would mean that *M. oleifera* grown in one area could be very different in applied properties as compared to those in another area. Therefore, the “moringa” studied by one research group sampling their local material could be very different from the “moringa” studied by another. Even more significantly, the moringa implemented in applications such as humanitarian nutritional interventions or for use in pharmacological clinical studies or nutraceutical applications might not offer the benefits intended. This “local adaptation” scenario seems plausible given the very wide cultivated range of *M. oleifera*.

The other possibility is that the M. oleifera material obtained from sampling in different parts of the tropical zone is more or less genetically similar throughout the world. This “low diversity domesticated” situation is plausible given that most *M. oleifera* material worldwide is used outside of its center of origin, India. Presumably the highest genetic diversity, and therefore diversity in applied properties, is found in India. From this center of diversity, movement around the world has plausibly led to successive losses of genetic diversity with each transference to a new location. Even within India, all of the true *M. oleifera* is domesticated. Domestication frequently involves a reduction in genetic diversity, often a very drastic one. Therefore, even within India, even though we still expect *M. oleifera* to have its highest genetic diversity there, the *M. oleifera* germplasm there almost certainly represents a significant reduction as compared to its as-yet unidentified wild progenitor. From this point of view, it is plausible that genetic variation in *M. oleifera* cultivated worldwide is sufficiently low that its applied properties are comparable wherever the plant is sampled in the world.

It is essential for applied studies in *Moringa* to distinguish between the “local adapatation” and “low diversity domesticated” situations because of their very different implications for the use and management of *Moringa* germplasm resources. If populations worldwide are highly distinct from one another in their properties of interest, then this requires careful screening of global germplasm in order to identify the variants with the most suitable characteristics for a given application. On the other hand, finding that worldwide variants are more or less comparable with regard to a given application makes implementation of *M. oleifera* much easier. It means that, for a given application, locally available plant material will be suitable for a given use, obviating the need for obtaining specific variants from other parts of the world, often at significant expense and bureaucracy.

Herein, we provide an example of how this issue can be addressed with regard to antioxidant effects with a common garden trial of *M. oleifera* material from diverse global provenances. One of the most important applications of *M. oleifera* worldwide is as dietary supplements and “nutraceutical” foods in managing chronic pre-pathological conditions such as hypercholesterolemia, insulin resistance, and inflammation. These conditions, whose onset is exacerbated by high levels of reactive oxygen species, have been shown to be reduced by the antioxidative effect of flavonoids and other glycosides present in moringa extracts [13,14,15,16]. Phenolic acids (e.g., chlorogenic acid and ferulic acid) are also present at moderate concentrations in extracts of *Moringa oleifera* leaves. These molecules contribute effectively as primary antioxidants, for example, by inactivating lipid free radicals or by acting in the prevention of the decomposition of hydroperoxides into free radicals [17,18,19]. Oxidative insult causes an imbalance in cellular redox homeostasis which, if not correctly counterbalanced, causes cellular damage that can lead to a pathological state [20].

In an effort to prevent or contain such harmful imbalances, a considerable amount of research has focused on testing nutritional strategies, especially for tissues particularly exposed to oxidative stress such as skeletal muscle. This research aims to demonstrate that supplementation with these extracts is effective in improving physical capabilities, such as a reduction in fatigue and increased exercise endurance [21,22,23,24]. In physiological conditions, ROS are maintained at low levels by several types of antioxidants. Among them, an important role is played by endogenous antioxidant enzymes (e.g., superoxide dismutase (SOD), catalase (CAT), glutathione peroxidase (GPx), glutathione transferase (GST)) and endogenous antioxidant molecules (mainly by the thiol system, glutathione (GSH), and thioredoxin (Trx)). In this context, manipulating concentrations of dietary antioxidants (e.g., vitamins, polyphenols, and flavonoids) provides a readily implemented means of intervention [25,26,27,28,29,30]. During intense or unaccustomed exercise, the high production of ROS ultimately leads to oxidative stress that causes myofibril damage with the consequent development of fatigue, a phenomenon that many sports practitioners experience and that leads to a deterioration in exercise performance. Muscle damage is evidenced by an increase in biomarkers of oxidation in both skeletal muscle and the blood [31,32]. Recently, we demonstrated that *Moringa oleifera* leaf extract (MOLE) improved oxidative capacity in C2C12 myotubes via the activation of the SIRT1-PPAR [33] and showed a beneficial effect on the antioxidant system of skeletal muscle cells through the induction of the nuclear factor erythroid 2-related factor (Nrf2) and its target gene heme oxygenase-1 (HO-1) pathway [34], restoring the redox status (total free thiols, Trx, and GSH/GSSG ratio) and increasing the antioxidant enzymatic system (CAT, SOD, GPx, and GST), thereby significantly reducing the thiobarbituric acid reactive substance (TBAR) and carbonylated protein (PrCAR) levels [35].

Given the very broad importance of ameliorating exercise-induced muscle damage, the use of *M. oleifera* as a nutritional supplement in counteracting the negative effects of ROS in muscle cells provides an excellent study system for evaluating variation associated with geographical provenance. Research to date largely consists of geographically limited studies that only examine materials available locally [36]. These practices make it unclear as to whether moringa samples from one area are superior to another, which would require identifying superior variants and distributing them globally. Alternatively, finding that globally cultivated moringa material is essentially functionally equivalent means that users can simply sample material available locally.

Here, we used ten samples of *M. oleifera* from a wide range of geographical provenances to test whether or not their levels of metabolite variation are very high or whether the samples are reasonably similar in antioxidant performance. The provenances examined included India and the Philippines in the area of traditional ancient *M. oleifera* cultivation. In Africa, we included Kenya, South Africa, and Madagascar, all areas with documented immigration from different parts of India. In the tropical Pacific region, we sampled Fiji and Réunion Island, also areas within the Indian diaspora. Finally, in the New World, we sampled *M. oleifera* grown on the Pacific coast of Mexico and the Caribbean coast of Colombia. Colonial records suggest that *M. oleifera* reached the New World on Spanish galleons along the Manila-Acapulco route. We grew these samples together in a common garden; finding that samples differ markedly in their properties despite growing under identical conditions would be consistent with the “local adaptation” scenario. To reflect the sorts of materials that are commercially available and often used in nutritional applications, we also included a purchased, commercial *Moringa oleifera* leaf powder (PureBodhi Nutraceuticals Ltd., London, UK) for comparison.

Thus, in this study, a metabolomics approach using LC-MS zenoTOF was adopted to facilitate the metabolite profiling of MOLE extracts, followed by Principal Component Analysis (PCA) to highlight correlations and discriminate samples of different geographical origins. Metabolomics is regarded as the main strategy for studying large numbers of samples. It is usefully coupled with LC-MS/MS techniques, which provide rapid and convenient methods for the simultaneous analysis of metabolite fingerprinting and quantification [37,38,39].

Subsequently, the antioxidant capacity of the leaf samples was evaluated using the Total Antioxidant Capacity assay.

Finally, the different MOLE extracts were compared with one another for their antioxidant activity on C2C12 myotubes subjected to an oxidative insult. Hydrogen peroxide (H_2_O_2_) was added to myotubes after pretreatment with different extracts and cell viability was assayed.

## 2. Results

### 2.1. Cultivation of Moringa oleifera Trees and Leaf Sampling

We brought together several accessions of *Moringa oleifera* from 4 continents and 9 countries (Table 1, Figure 2) and grew them together in a common garden, the International *Moringa* Germplasm Collection (Jalisco, Mexico) until reaching mature size. Leaf samples were then collected and immediately dried in a large quantity of silica gel desiccant, following previous work that shows that silica gel drying provides excellent preservation of glucosinolates and myrosinase [9], during the dry season and stored in a dry environment until used to obtain the extracts.

### 2.2. LC-MSMS ZenoTOF MOLE Qualitative Profiling

Qualitative profiling was carried out on a ZenoTOF 7600 system interfaced with ultra-high performance liquid chromatography (SCIEX, Framingham, MA, USA).

Chromatographic separation was optimized using a Kintex F5 column (150 mm × 2.1 mm, 2.6 µm, Phenomenex, Torrance, CA, USA). The optimized gradient was as follows: at 0 min, 0.2% buffer A; at 15 min, 95% buffer A. Buffer A was water acidified with 0.1% formic acid, while buffer B was acetonitrile (ACN) acidified with 0.1% formic acid. The flow rate was 0.2 mL/min, and the column oven was maintained at 40 °C. Five microliters of each sample were injected and acquired using Information-Dependent Acquisition (IDA), while for digital fingerprinting, the preferred acquisition mode was SWATH data-independent acquisition (DIA) analysis. The source parameters were as follows: CUR = 30 psi, CAD = 7, IS = −4500 V, TEM = 400 °C, GS1 = 50 psi and GS2 = 50 psi. Data were processed using SCIEX OS software, version 3.3 (SCIEX, Framingham, MA, USA). The SCIEX Natural Products 1.6 Library (SCIEX, Framingham, MA, USA) and NIST 2017 library were used to search compound spectra saved to various databases. Univariate and multivariate statistical analyses were performed using MarkerView software, version 1.3.1 (SCIEX, Framingham, MA, USA).

At the first stage of this study, aiming for a comprehensive metabolomic analysis, Principal Component Analysis (PCA) with a non-targeted approach was used. Subsequently, a Partial Least Squares-Discriminant Analysis (PLS-DA) was performed to visualize the discrimination of various *Moringa* samples based on their MS signals. The results are shown in Figure 3. The score plot (Figure 3, left) clearly shows that UK samples (in the red circle) are located far away from the others at the most highly negative D1 and D2 values. Similarly, the samples from Colombia (yellow circle) were discriminated from the others, with them being located at the most positive D1 values. Another cluster was located at the mostly highly negative D2 values (after the UK samples), made up of samples from Réunion and Fiji (blue circle). The loading plot (Figure 3, right) showed the features responsible for sample discrimination. As examples, we focus on features *m*/*z* 353 (1078) (Figure 4), feature *m*/*z* 609 (Figure 5), and feature *m*/*z* 570 (Figure 5). Surprisingly, by extracting the feature at *m*/*z* 353, we found 2 peaks at different retention times (Figure 4 XIC panels; 4.9 and 5.4 min) showing different trends among the *Moringa* samples. UK samples overexpressed the feature at 5.4 min and had an opposite trend for the feature at 4.9 min. TOF-MS panels showed the same accurate mass for both features, while the TOF MS/MS panels highlighted a different fragmentation pattern. According to library matching (SciexOS software version 3.3—SCIEX Natural Products 1.6 Library (SCIEX) and NIST 2017 library), these isomers were identified as cryptocholorogenic acid (Rt 5.4 min) and neochlorogenic acid (Rt 5 min).

With regard to the feature at *m*/*z* 609, the plot profile (Figure 5, top panel) shows overexpression only in the Colombia samples. Library matching points to this feature being rutin.

Instead, the feature at *m*/*z* 570 exhibited a higher trend in the Kenya and Colombia samples, followed by those from India and Mexico. This feature did not match with any library entries, but TOF MS and TOF MS/MS spectra in accurate mass suggest glucomoringin. For confirmation, we used Formula Finder and Chemspider tools in SciexOS software (SciexOS software version 3.3). Figure 6 shows the Chemspider results, which confirm the matching with glucomoringin.

With the aim to deeper profile the MOLE extracts, both workflows, DDA and DIA (ZenoSWATH), were used. ZenoSWATH allowed for more confident identification in the case of very low abundance secondary metabolites. The detection of very low-abundance metabolites is enhanced by searching for precursor ions and subsequently detecting their diagnostic fragment ions with accurate mass measurements. An example is shown in Figure 7. By monitoring the transition from *m*/*z* 577 to *m*/*z* 289, we were able to detect and confidently identify three isomers of procyanidns with retention times of 5.4, 5.7, and 6.2 min.

In this way, we were able to putatively identify around 77 metabolites within MOLE samples with different percentages (Table 2 and Table 3).

### 2.3. Cell Viability

Myotube viability was assessed using the methylthiazolyldiphenyl-tetrazolium bromide (MTT) assay [40].

Briefly, C2C12 myotubes were treated with different leaf extract stock solution dilutions (1/100 working solution) or vehicle (ethanol) in culture media for 24 h. During the last hour of treatment, a sample treated with 1 mM H_2_O_2_ was also tested (Figure 8 upper panel). No statistically significant differences were observed in myotubes supplemented with MOLE. However, a 28% reduction in cell viability was observed in myotubes treated with H_2_O_2_ (*p* < 0.01, Figure 8, upper panel).

To assess the protective effect of MOLE, C2C12 myotubes were treated with dilutions of various MOLE stock solution (1/100 working solution) or vehicle (ethanol) in culture media for 24 h. After incubation, hydrogen peroxide (1 mM) was added to the MOLE pre-treated samples for an additional hour, followed by an MTT assay (Figure 8, lower panel).

We found that myotubes pre-treated with MOLE and subsequently exposed to hydrogen peroxide exhibited significantly greater cell viability, with increases ranging from 12% (Kenya, *p* < 0.05) to 16% (South Africa, *p* < 0.05) compared to H_2_O_2_-treated controls for all samples tested (Figure 8, lower panel).

Cell survival was also measured using a 3-(4,5-dimethylthiazol-1)-5-(3-carboxymeth-oxyphenyl)-2H-tetrazolium (MTS) assay (Promega, Madison, WI, USA) [41]. The results coincide with the MTT test in showing greater proportions of cell viability in MOLE-treated samples (Appendix A).

### 2.4. Evaluation of MOLE Trolox Equivalents Antioxidant Capacity

Total antioxidant capacity (TAC) was performed spectrophotometrically using the Trolox equivalents antioxidant capacity assay (Sigma-Aldrich Chemical, St. Louis, MO, USA). The analysis of the total antioxidant capacity of the various MOLE samples showed no statistically significant differences among them (*p* > 0.05). MOLE-India and MOLE-UK (commercial) had the highest values (0.29 ± 0.02 and 0.30 ± 0.01 Trolox equivalents millimolar/mg, respectively). MOLE-Fiji and MOLE-Réunion had the lowest values (0.22 ± 0.01 and 0.23 ± 0.01 Trolox eq. millimolar/mg, respectively, Figure 9).

## 3. Discussion

*Moringa oleifera* Lam. offers a wealth of properties with significant implications for both therapeutic and nutritional applications. Different parts of the plant, such as the leaves, seeds, and roots, are used to treat various pathological conditions. Moreover, it supports cardiovascular health, regulates blood glucose levels, and exhibits antioxidant, anti-inflammatory, and potential anti-cancer properties [1,2,3,4,7,8,13,14,15,16,17,18,19]. Moringa leaves are highly nutritious, and in low-income countries they are very valuable in combating food and nutrition insecurity. In fact, given its multiple therapeutic activities, *Moringa oleifera* is known as a “miracle tree”. Incorporating Moringa oleifera into one’s everyday diet is an easy way to benefit from its therapeutic and nutritional properties. It can be eaten fresh, i.e., in salads, or dried (moringa powder) and added, i.e., to yogurt or juices or herbal tea. Otherwise, it can be used in seeds or oil. Another convenient option is taking moringa capsules or supplements. Skeletal muscle is a tissue that is often exposed to pro-oxidizing conditions due to its high oxygen consumption rates. In our previous study, we reported that MOLE pretreatment had a beneficial effect on the antioxidant system of skeletal muscle cells under the stressful conditions of an oxidizing environment [33,34,35].

In the present work, we have found that *Moringa oleifera* leaves obtained from different specimens originating from different regions of the world but grown in the same environmental conditions proved for all practical purposes equally effective in counteracting the harmful effects of the oxidative insult induced by hydrogen peroxide in C2C12 myotubes and this is despite the difference in the percentage of bioactive molecules present in the extracts.

These results are highly encouraging because they suggest that, at least for the present application, *Moringa oleifera* germplasm can be used essentially regardless of geographical provenance. In our study, we brought together accessions of *Moringa oleifera* from 4 continents and 9 countries and grew them together in a common garden, the International Moringa Germplasm Collection (Jalisco, Mexico) until reaching mature size. After that, the leaves were obtained and dried, and ethanolic extraction was performed.

We did detect some variation in phytochemical profiles across the MOLE samples. The highly sensitive ZenoTOF 7600 mass spectrometer enabled the in-depth qualitative profiling of the MOLE extracts. The integration of both Zeno-DDA and Zeno-SWATH approaches significantly enhanced the accuracy of bioactive molecule detection in MOLE extracts. The metabolomics analysis identified the presence of glucosinolates, flavonoids, phenolic acids, and other metabolites (such as aminoacids, vitamins, and fatty acids) in all samples, albeit in varying proportions. All samples exhibited the highest percentage of glucosinolates, except for the samples from Kenya, which had the highest percentage in the “other compounds” group. Among the glucosinolates, the sample from Colombia had the highest content, with a percentage of 50.8 of the total area, followed by India (49.9%) and Mexico (48.2%). With regard to flavonoids, the Madagascar extract had the highest percentage, at 30%, followed by the samples from South Africa and Colombia. With regard to phenolic acids, Fiji and Réunion were the richest while for the “other compounds” group, the richest were Kenja and Fiji.

Interestingly, in vitro analysis of the total antioxidant capacity of the various extracts revealed similar activity across all tested samples despite some variation in phytochemical profiles. This similarity of the different extracts was further supported by their effectiveness in mitigating oxidative damage induced by H_2_O_2_ in a cell culture model. All extracts demonstrated comparable efficacy in restoring C2C12 myotubes following oxidative stress when pretreated with MOLE. This uniform response is likely attributable to the collective antioxidant properties of the bioactive molecules present in Brassicalean extracts, and *Moringa oleifera* in particular, as documented in previous studies [42,43,44,45,46,47,48]. Thus, our results showed that even though there was some statistically significant variation in the phytochemical profiles, this variation was in a relatively narrow absolute range, and was not sufficient to provoke differences in antioxidant potential.

Isothiocyanates, polyphenols, flavonoids, and phenolic acids are known to act as antioxidants, either inactivating lipid free radicals or preventing hyperperoxide decomposition [49,50,51]. Glucosinolates (GSLs) are sulfur-containing glucosidic compounds typical of the order Capparales (e.g., Brassicaceae, Capparaceae, Caricaceae, etc.) that are known for their health-promoting and antioxidative properties mediated by their metabolites [52,53,54,55,56]. Upon chewing or mechanical processing, glucosinolates are hydrolyzed into isothiocyanates due to myrosinase enzyme activity. Over 100 isothiocyanates have been identified, including benzyl isothiocyanate, phenyl isothiocyanate, and sulforaphane [57], which are potent activators of antioxidant defense pathways, supporting mitochondrial function and maintaining protein integrity under oxidative stress. These antioxidant properties are primarily mediated through the Nrf2-dependent antioxidant cellular response. Isothiocyanates also exhibit Nrf2-independent effects, such as inhibition of mitochondrial fission and modulation of the mTOR pathway [58]. In individuals undergoing oxidative stress induced by physical exercise, sulforaphane, the most extensively studied isothiocyanate, has demonstrated efficacy in reducing muscle damage and inflammation [59] and alleviating muscle soreness by upregulating Nrf2-target NQO1 expression [60].

A glucosinolate-rich extract effectively enhanced the antioxidant defense system by upregulating Nrf2-mediated gene induction, including GCLC, NQO1, and HO-1 mRNA levels, as well as increasing HO-1 protein levels. Additionally, it activated the p38 MAPK signaling pathway [61], known for its role in regulating Nrf2 phosphorylation and nuclear translocation [62,63].

*Moringa oleifera* leaf extracts are rich sources of polyphenols, a diverse group of naturally occurring compounds usually abundant in plants, characterized by multiple phenol groups and encompassing various chemical structures and biological functions [64]. Polyphenols, including subclasses like flavonoids, flavanols, and phenolic acids [65,66], are well-known for their antioxidant properties, which involve scavenging and neutralizing free radicals, mitigating oxidative stress and cellular damage induced by reactive oxygen species (ROS) [67]. Flavonols such as quercetin and kaempferol are renowned for their potent antioxidant activity, inhibiting lipid peroxidation, and enhancing endogenous antioxidant defenses through the activation of enzymes like SOD and CAT [68]. Quercetin has been shown to promote mitochondrial biogenesis in skeletal muscles, thereby improving mitochondrial function, protein content, enzyme activity, and respiratory function [69,70,71,72,73]. Furthermore, quercetin protects myotubes against TNF-induced muscle atrophy under obese conditions by inducing Nrf2-mediated HO-1 induction while inhibiting NF-kB activation [74]. Catechin flavonoids, including epicatechin and epigallocatechin, are recognized for their antioxidant properties [75], which involve scavenging free radicals, modulating cellular signaling pathways, and enhancing the activity of endogenous antioxidants [76,77,78,79].

Phenolic acids, such as hydroxybenzoic acids (e.g., gallic acid) and hydroxycinnamic acids (e.g., caffeic acid), are also present in leaf extracts. These compounds represent another important group of polyphenols [80] with antioxidative properties that contribute to cellular defense against oxidative stress. Gallic acid scavenges free radicals, inhibits lipid peroxidation, and preserves cellular integrity by modulating antioxidant enzyme activity [81]. Similarly, caffeic acid scavenges free radicals, chelates metal ions involved in oxidative processes, and regulates gene expression related to antioxidant defenses [82].

Polyphenols act as antioxidants in various ways. For example, they modulate the activity of endogenous antioxidant enzymes such as SOD, CAT, and GPx, which are crucial in neutralizing free radicals and ROS, thereby reinforcing the cellular antioxidant defense system [83,84]. Additionally, polyphenols directly scavenge free radicals, reducing their reactivity and potential to cause cellular damage, and mitigate oxidative damage resulting from inflammatory processes [83]. Polyphenols are also effective in regenerating other antioxidants, such as vitamins C and E, enhancing their antioxidative efficacy within the cellular environment [85]. Moreover, they activate specific cellular defense mechanisms and promote the repair of damaged molecules, thereby enhancing cellular resilience against oxidative damage [86].

The disruption of cellular antioxidant system homeostasis is a key feature of oxidative stress. To address the growing demand for nutritional interventions to counteract oxidative stress, research has increasingly focused on nutritional strategies aimed at enhancing physical capabilities, such as reducing fatigue and increasing exercise endurance [87,88,89,90].

Skeletal muscle tissue is particularly susceptible to oxidative stress. Muscle contractions during physical exercise, especially intense or unaccustomed activities, are typically accompanied by high ROS production, leading to oxidative stress and potential myofibral damage. This damage is evidenced by increased biomarkers of oxidation in both skeletal muscle and the blood [31,32]. Furthermore, oxidative stress is a key factor in the development of fatigue, a common experience among athletes that leads to a decline in exercise performance. To preserve muscle function and protect myotubes from excessive ROS exposure, the use of antioxidants is a common strategy. Appropriate antioxidant use has been shown to be beneficial in balancing the ratio between oxidants and antioxidants in most physiopathological conditions [87,91,92].

In this context, the use of *Moringa oleifera* extracts on muscle cell models has yielded encouraging results in counteracting oxidative stress. Our group has previously demonstrated that MOLE exhibits a dose-dependent total antioxidant capacity in a cell-free system, indicating that its efficacy depends on the concentration of antioxidant molecules present in the mixture [33,35]. Interestingly, treatment with MOLE activated oxidative metabolism through the SIRT1-PPAR pathway and the Nrf2 pathway, along with its target gene HO-1, both of which are regulators of cellular resistance to oxidants. MOLE also improved glutathione redox homeostasis and increased antioxidant enzymatic activities in C2C12 myotubes [33,34]. Moreover, a pre-supplementation strategy with MOLE showed a significant protective effect on C2C12 myotubes against oxidative insult induced by acute H_2_O_2_ treatment [35].

Metabolomics analysis of our samples highlighted a similar profile of the identified metabolites. However, differences between the different extracts were found when considering the relative percentages of the different groups of molecules (glucosinolates/isothiocyanates, flavonoids, and phenolic acids) in the MOLE samples.

Furthermore, it must be considered that in MOLE, there are many bioactive molecules with antioxidant action per se; however, it must be taken into consideration that the biological effect of the extracts derives not from only a single component but is probably due to a synergistic effect of the mixture of all of the bioactive molecules present [33,34,35]. Among them, the most representative molecules in our samples are the glucosinolates glucomoringin and 4-O-acetylrhamnopyranosyloxybenzyl-GS; the flavonoids isoquercitrin, astragalin, and rutin; the phenolic acid chlorogenic acid; and among the lipids, omega-3 alpha-linolenic acid. From the literature, it has been shown that these molecules can contribute per se [93,94,95] or with a synergistic effect [33,34,35] to the antioxidant protective action. Here, we evaluated the synergic effects of the *Moringa oleifera* leaf extracts.

The results obtained in this work demonstrate that different *Moringa oleifera* specimens, despite being cultivated under the same environmental conditions, produce extracts with varying qualitative characteristics but similar biological effects. This is particularly important as these qualitative differences are entirely due to heritable variation among individuals from different geographical locations, rather than cultivation conditions. Consequently, this finding enhances the validity of using extracts from this plant for nutritional purposes wherever it is grown.

Future studies are warranted on the evaluation of the effects of environmental variation on the growth of plants coming from the same region and how these variations may possibly influence the quality of the biomolecules present in the different parts of the plant and in particular in the leaves.

## 4. Materials and Methods

All chemical reagents, unless otherwise specified, were purchased from Sigma-Aldrich Chemical (Sigma-Aldrich Chemical, St. Louis, MO, USA). A schematic representation of the workflow is shown in Figure 10.

### 4.1. Cultivation of Moringa oleifera Trees and Leaf Sampling

The plants were grown in a common garden located at the International *Moringa* Germplasm Collection near the Chamela Biological Station of the Universidad Nacional Autónoma de México, situated on the Mexican Pacific coast in Jalisco State. This region experiences a tropical monsoonal climate characterized by a rainy season from July to October, interspersed with a prolonged dry season. The average annual rainfall is 752 ± 256 mm, predominantly occurring during a few significant events. The mean annual temperature is 24.9 °C, ranging from 14.8 to 32 °C [96,97]. The garden features a consistent base soil composed of decomposed granodiorites originating from the Vallarta Batholith [98].

Our samples describe much of the variation that is well-documented within *M. oleifera*, e.g., spanning variants with short fruits with few, large seeds to those with long fruits and many small seeds; leaves with greater or lesser degrees of red pigment on the rachis; or variation in flower color, from white to cream [99,100,101]. The individuals we studied here have been previously examined for variables such as protein content and glucosinolate profiles, and they do show some variation [10], though accessions of *M. oleifera* are much more similar to one another than they are to other Moringa species [9,102].

The uppermost fully expanded leaves were collected from randomly selected branches of healthy plants and promptly dried using ample silica gel desiccant. Once completely dehydrated, the leaves were divided into sample portions of a few grams and then stored for subsequent extraction.

### 4.2. MOLE: Ethanolic Extract of Moringa oleifera Leaves

The dried leaves from different samples (Table 1) were finely chopped and then 1 g of leaf powder was used. For comparison, commercial *Moringa oleifera* leaf powder (PureBodhi Nutraceuticals Ltd., London, UK) was also tested. Briefly, 1 g of leaf powder was dissolved in 10 mL of ethanol (100%) and then sonicated (Vibra-Cell CV 18 SONICS VX 11, Sonics & Materials, Newtown, CT, USA) twice for 10 min at +4 °C. The resulting extract was centrifuged (2000× *g* for 10 min at +4 °C), collected, and stored at −20 °C (stock solution corresponding to 15 mg/mL of dried leaves).

### 4.3. LC-MSMS SCIEX ZenoTOF 7600 System MOLE Qualitative Profiling

Qualitative profiling of the MOLE extracts was conducted using ultra-high performance liquid chromatography–quadrupole time-of-flight mass spectrometry (UHPLC/QTOF- MS) on a SCIEX X500B system equipped with a ZenoTOF 7600 mass spectrometer (AB SCIEX GmbH, Landwehrstraße 54, Darmstadt, Germany). The instrument utilized a high-resolution QTOF electrospray ion source operated in negative ion mode, following established protocols [33]. The sample’s digital fingerprint was characterized using SWATH analysis.

The data obtained were processed using SciexOS Software 3.3 (AB SCIEX GmbH, Landwehrstraße 54, Darmstadt, Germany), and the SCIEX Natural Products 2.1 Library (AB SCIEX GmbH, Landwehrstraße 54, Darmstadt, Germany) was used to search compound spectra databases.

### 4.4. Cell Cultures

C2C12 myoblasts (ATCC, Manassas, VA, USA) were cultured following established protocols. Preconfluent cells (85% confluency) were induced to differentiate by lowering the FBS to 2% in a culture medium. Cell differentiation was monitored using microscopy and assessed using myogenin and MHC expression with Western blot analysis [103].

C2C12 myotubes were treated with MOLE working solutions (1/100 dilution of stock solutions) or vehicle-only (ethanol) in culture medium for 24 h to assess cytotoxicity. The ethanol concentration in the working solutions (0.1%, v/v) did not affect the myotubes. During the final hour of treatment, a sample exposed solely to 1 mM H_2_O_2_ was tested. Cell viability was assessed with the methyl-thiazolyl-diphenyl-tetrazolium bromide (MTT) assay [104].

To verify the protective effect of different *Moringa oleifera* leaf extracts, myotubes were treated with MOLE solutions or vehicle-only (methanol) in culture medium for 24 h. Subsequently, hydrogen peroxide (1 mM) was added to samples pre-treated with vehicle or MOLE for a further hour. MTT assays were then performed, and the samples were prepared for biochemical analysis. The cells were trypsinized, collected, and centrifuged at 1200× *g* for 10 min at room temperature. For comparison, cell survival was also measured using a 3-(4,5-dimethylthiazol-1)-5-(3-carboxymeth-oxyphenyl)-2H-tetrazolium (MTS) assay (Promega, Madison, WI, USA) [41] (Appendix A). After gentle lysis, the lysate was used for biochemical analysis or tested for protein content using the Bradford method (Sigma-Aldrich, St. Louis, MO, USA).

### 4.5. TAC: Trolox Equivalents Antioxidant Capacity

The total antioxidant capacity (TAC) was performed spectrophotometrically using the Trolox equivalents antioxidant capacity assay [105]. This assay evaluates the ability of cell lysates to prevent ABTS+ radical formation in ABTS-metMyoPBS buffer after the addition of H_2_O_2_ (450 μM) compared to the vitamin E analog Trolox standards. The variation in absorbance detected at 734 nm was compared to those obtained using Trolox standards (0.125–2.0 mM) and expressed as micromoles per milligram of protein (µmol/mg protein) tested.

To verify the efficiency of the extraction method, 10 μL of different MOLE stock solution dilutions (0.015, 0.075, 0.15, and 1.5 mg/mL of dried powder) was tested and the antioxidant capacity observed was comparable with that already reported [33,34].

### 4.6. Statistical Analysis

The distribution of the data was evaluated using the Kolmogorov–Smirnov test. All of the data are expressed as the means ± S.D. of three independent experiments, each performed in triplicate. One-way ANOVA for repeated measures and Bonferroni post-hoc analyses were performed to test for significant differences among groups for each variable tested. Statistical significance was set at *p* < 0.05. Statistical analyses were performed using SPSS for Windows (Version 17.0; SPSS Inc., Chicago, IL, USA). Comparisons between untreated controls and control vehicles showed no statistical differences for all variables tested [33,34].

## 5. Conclusions

In conclusion, we provide very encouraging results that *Moringa oleifera* leaf extract is effective in reducing the damaging effect of oxidative insult in C2C12 myotubes irrespective of geographical provenance. These findings are of particular importance because they suggest that the use of Moringa for its therapeutic benefits can proceed without the need for the lengthy and complex global exchange of materials between regions. To date, because studies tend to examine only locally available germplasm, it has remained unclear as to whether the material from one area differs significantly from that available in another. In the context of the prevention of oxidative stress, avoiding skeletal muscle cellular damage via dietary antioxidants promises to be a low-cost, convenient, and highly effective strategy. Our results suggest that given the multitude of bioactive molecules present, although in different percentages, whatever *Moringa oleifera* material is available locally can be used both as nutritional support and as a useful agent to counteract oxidative stress and/or to assist traditional medicine in the treatment of various pathologies.

## Figures and Tables

**Figure 1 ijms-25-08109-f001:**
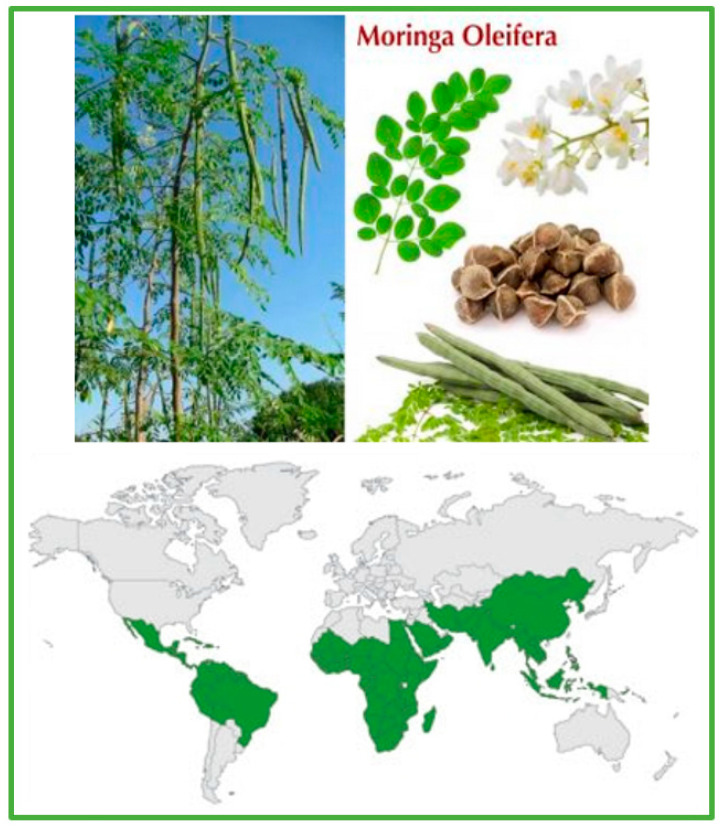
Images of the habit (green on the map), flowers, leaves, fruits, seeds and areas in which the “miracle tree” *Moringa oleifera* Lam. is recorded as being cultivated (data from https://www.gbif.org/ accessed on 17 January 2024).

**Figure 2 ijms-25-08109-f002:**
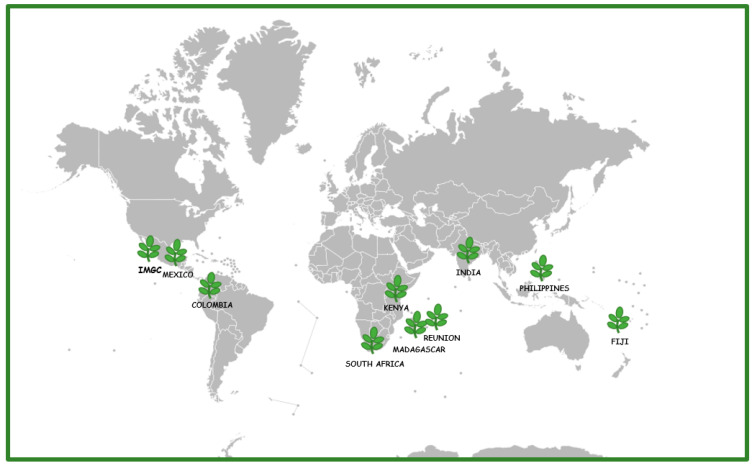
Origin of the different individuals of *Moringa oleifera* Lam. grown in the IMGC (International *Moringa* Germplasm Collection, Mexico).

**Figure 3 ijms-25-08109-f003:**
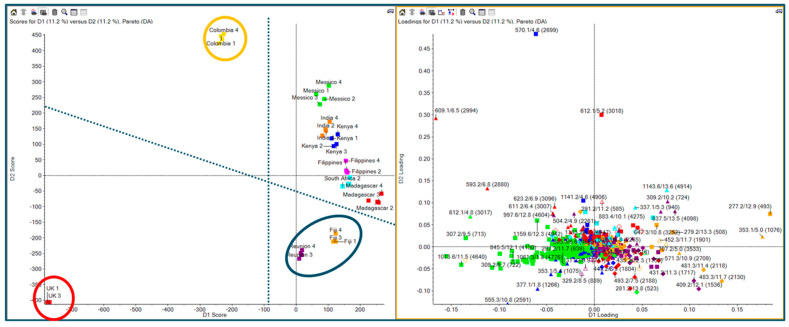
PCA supervide analysis (PLSDA) score plot (**left**) loading plot (**right**).

**Figure 4 ijms-25-08109-f004:**
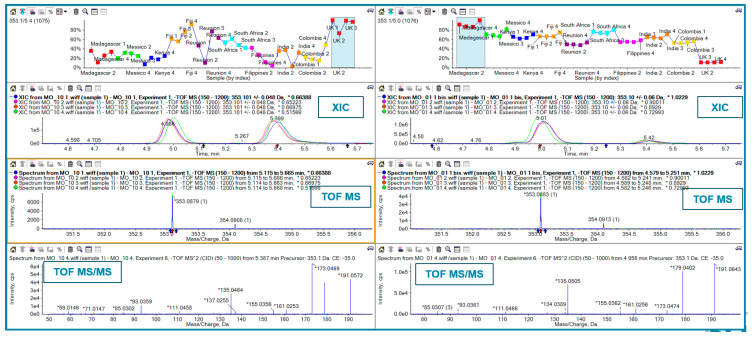
Plot profiles, XICs, TOF MS, and TOF MS/MS for feature *m*/*z* 353. In the XIC panels red arrow shows the interested chromatographic peak and blue arrows the isolation width. In the TOF MS panels, red arrow shows the selected precursor ion and blue arrows the isolation width.

**Figure 5 ijms-25-08109-f005:**
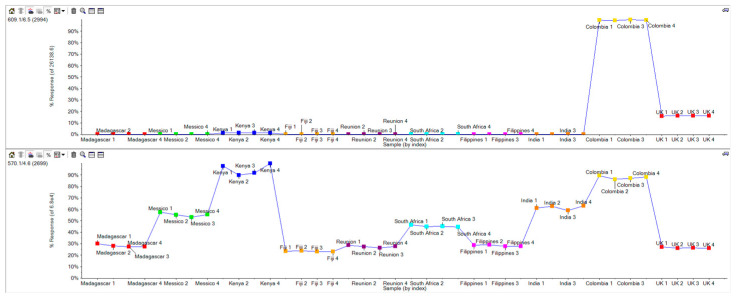
Plot profiles for features *m*/*z* 609 and 570.

**Figure 6 ijms-25-08109-f006:**
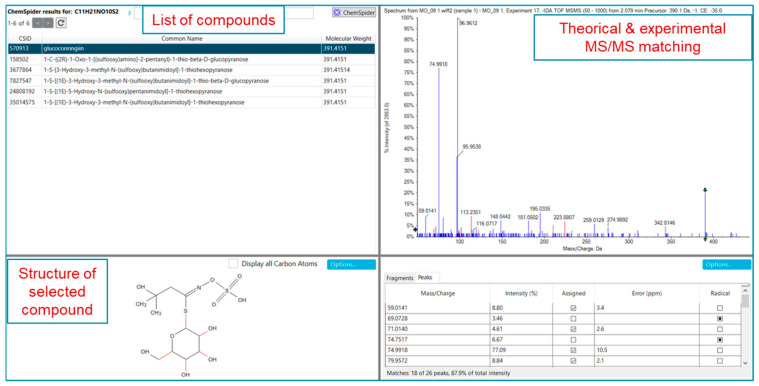
SciexOS software analysis (SciexOS software version 3.3). Green arrows indicate the selected precursor ion.

**Figure 7 ijms-25-08109-f007:**
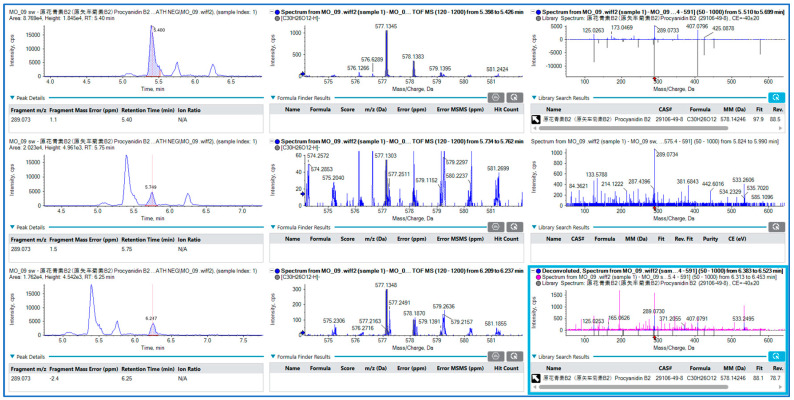
Extracted ion chromatograms (XICs) of the fragment at *m*/*z* 289.073, along with TOF MS and ZenoSWATH MS/MS of procyanidin isomers. Red arrows indicate the selected diagnostic fragment ion.

**Figure 8 ijms-25-08109-f008:**
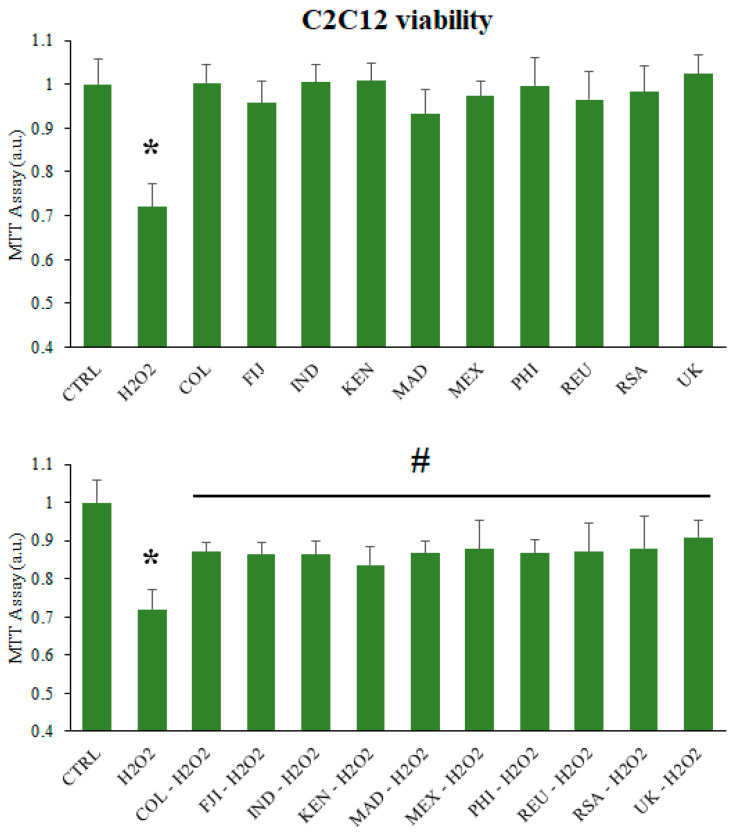
MTT assay. (**Upper Panel**) C2C12 myotubes were treated with various dilutions of MOLE stock solution (1/100 working solution) or vehicle (ethanol) in culture media for 24 h. During the final hour of treatment, a sample treated solely with 1 mM H_2_O_2_ was tested. (**Lower Panel**) C2C12 myotubes were treated with MOLE or vehicle (ethanol) in culture media for 24 h. Hydrogen peroxide (1 mM) was then added to the MOLE-pre-treated samples for an additional hour. Cell viability was assessed using the MTT assay. Data are presented as the mean ± S.D. of three experiments, each performed in triplicate. * *p* < 0.01 vs. CTRL; # *p* < 0.05 vs. H_2_O_2_.

**Figure 9 ijms-25-08109-f009:**
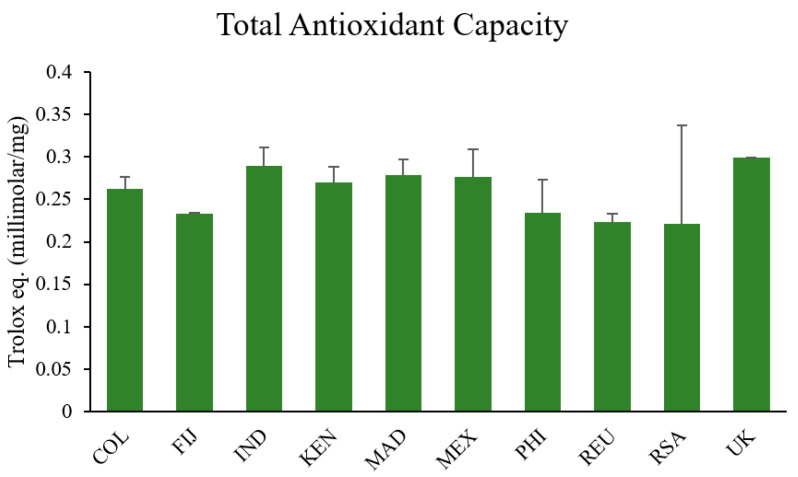
Total antioxidant capacity (TAC). Measurement of total antioxidant capacity (TAC) was tested in different MOLE stock solution dilutions (1/100 working solution). Data presented are the mean ± S.D. of three experiments.

**Figure 10 ijms-25-08109-f010:**
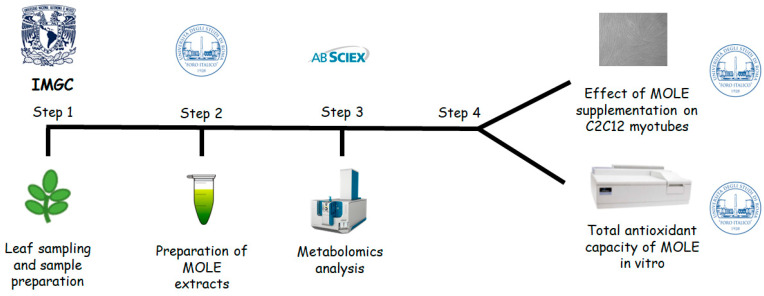
Schematic representation of the workflow.

**Table 1 ijms-25-08109-t001:** *Moringa oleifera* sampling.

IMGCC ^1^ Accession	Country of Origin	Locality
1	Madagascar	Tolagnaro
2	Mexico	Mérida, Yucatán
4	Kenya	Isiolo Town, Isiolo County
34	Fiji	Suva, Viti Levu
82	Réunion	Saint-Leu, Île de la Réunion
324	South Africa	Commercial nursery
362	Philippines	Mabini Town, Batangas Province
382	India	Jodhpur, Rajasthan
383	Colombia	Cartagena, Departamento de Bolívar

^1^ International *Moringa* Germplasm Collection accession number.

**Table 2 ijms-25-08109-t002:** Relative percentage of the presence of the different 77 metabolites identified in the different MOLE samples divided by groups of molecules (glucosinolates/isothiocyanates, flavonoids, phenolic acids, and others).

	Colombia	Fiji	India	Kenya	Madagascar	Mexico	Philippines	Reunion	South Africa	UK
**Glucosynolates/isothicianates %**										
hydroxybutyl (390.05_2.1)	0.05	0.01	0.01	0.10	0.24	0.02	0.01	0.02	0.01	0.00
Glucomoringin (570.1_4.6)	17.00	5.93	9.71	24.22	4.77	9.00	4.67	10.08	8.56	10.31
Glucosoonjnain (586.09_4.6)	0.06	0.02	0.03	0.07	0.02	0.03	0.01	0.03	0.02	0.03
4-O-acetylrhamnopyranosyloxybenzyl-GS 1 (612.11_5.2)	28.91	25.85	34.59	0.08	24.15	34.88	33.59	20.01	22.63	22.34
4-O-acetylrhamnopyranosyloxybenzyl-GS 2 (612.11_4.8)	3.18	2.99	3.58	0.01	1.91	2.90	2.81	3.14	2.43	6.10
4-O-acetylrhamnopyranosyloxybenzyl-GS 3 (612.11_4.9)	1.40	1.47	1.79	0.01	0.86	1.16	1.16	1.59	1.35	3.12
4-O-acetylglucopyranosyloxybenzyl-GS 1 (628.1_5.1)	0.10	0.06	0.11	0.00	0.07	0.10	0.10	0.05	0.06	0.10
4-O-acetylglucopyranosyloxybenzyl-GS 2 (628.1_5.2)	0.02	0.02	0.02	0.00	0.03	0.04	0.03	0.01	0.01	0.02
4-O-acetylglucopyranosyloxybenzyl-GS 3 (628.1_4.7)	0.02	0.01	0.01	0.00	0.01	0.01	0.01	0.01	0.01	0.03
4-O-acetylglucopyranosyloxybenzyl-GS 4 (628.1_4.8)	0.01	0.01	0.01	0.00	0.01	0.01	0.01	0.01	0.01	0.02
C22H41O15NS3 (654.16_6.1)	0.00	0.00	0.00	0.00	0.00	0.00	0.00	0.00	0.00	0.01
C22H41O15NS3 bis (654.15_6.1)	0.00	0.00	0.00	0.00	0.00	0.00	0.00	0.00	0.00	0.01
Glucoputranjivin (360.04_2.1)	0.01	0.00	0.01	0.01	0.01	0.01	0.00	0.00	0.00	0.00
Glucoconringiin (390.05_2.1)	0.05	0.01	0.01	0.10	0.24	0.02	0.01	0.02	0.01	0.00
Glucosinalbin1 (424.04_4.6)	0.01	0.00	0.00	0.01	0.00	0.01	0.00	0.00	0.00	0.01
Glucoalyssin (450.06_5.2)	0.00	0.00	0.00	0.00	0.00	0.00	0.00	0.00	0.00	0.00

	**Colombia**	**Fiji**	**India**	**Kenya**	**Madagascar**	**Mexico**	**Philippines**	**Reunion**	**South Africa**	**UK**
**Flavonoids %**										
Pinocembrin 1 (255.07_8.4)	0.01	0.00	0.00	0.02	0.02	0.01	0.01	0.00	0.01	0.03
Pinocembrin 2 (255.07_8.1)	0.01	0.00	0.00	0.02	0.01	0.01	0.01	0.01	0.01	0.02
Kaempferol (285.04_9)	0.01	0.02	0.05	0.02	0.06	0.01	0.05	0.03	0.03	0.04
Catechin (289.07_5.8)	0.15	0.01	0.01	0.11	0.01	0.03	0.07	0.00	0.02	0.00
Epicatechin (289.07_5.5)	0.01	0.00	0.00	0.01	0.00	0.00	0.00	0.00	0.00	0.00
Salidroside (299.11_5)	0.04	0.02	0.03	0.13	0.02	0.04	0.03	0.03	0.04	0.06
Quercetin (301.04_8.3)	0.03	0.07	0.01	0.03	0.08	0.04	0.02	0.06	0.04	0.04
Isorhamnetin (315.05_9.1)	0.01	0.04	0.02	0.03	0.04	0.04	0.01	0.05	0.03	0.02
Esculin (339.07_5.3)	0.06	0.06	0.05	0.06	0.02	0.04	0.05	0.13	0.05	0.12
Puerarin 1 (415.1_6.9)	0.01	0.01	0.00	0.02	0.02	0.01	0.01	0.01	0.01	0.01
Puerarin 2 (415.1_7.5)	0.01	0.01	0.00	0.02	0.03	0.00	0.01	0.01	0.01	0.01
Vitexin (431.1_6.4)	0.33	0.41	1.12	1.63	2.00	2.01	1.32	1.83	1.52	1.22
Isovitexin (431.1_6.6)	0.69	0.54	1.49	2.45	2.68	2.80	1.78	2.20	2.16	1.38
Genistin (431.1_7.6)	0.01	0.01	0.03	0.01	0.03	0.02	0.02	0.02	0.02	0.01
Astragalin (447.09_7)	2.03	2.50	8.97	2.67	8.98	0.95	6.48	2.41	6.52	4.50
orientin_2 (447.09_6.1)	0.17	0.10	0.14	0.23	0.51	0.13	0.31	0.37	0.33	0.16
Kaempferide 7-O-glucoside 1 (461.11_7.3)	0.02	0.03	0.03	0.04	0.03	0.01	0.05	0.03	0.03	0.07
Kaempferide 7-O-glucoside 2 (461.11_7.5)	0.00	0.01	0.01	0.01	0.01	0.00	0.01	0.01	0.01	0.01
Isoquercitrin (463.09_6.7)	6.67	7.65	2.67	7.14	8.62	5.43	3.15	6.87	7.29	6.30
Nepetin 7-glucoside (477.1_7.1)	0.75	0.63	0.42	0.74	1.28	0.35	0.17	0.86	0.95	0.62
Quercetin-O- ð› ½-D-glucose-acetate isomer 1 (505.1_6.8)	1.74	1.27	0.47	1.46	2.97	1.51	0.67	0.69	1.57	0.02
Quercetin-O- ð› ½âˆ’ D-glucose-acetate isomer 2 (505.1_7)	0.23	0.16	0.04	0.18	0.34	0.14	0.07	0.09	0.18	0.00
Quercetin-O- ð› ½-D-glucose-acetate isomer 3 (505.1_7.1)	0.10	0.19	0.03	0.14	0.26	0.09	0.08	0.18	0.16	0.49
Kaempferol 3-O-(3 â€² â€™,4 â€² â€™-di-O-acetyl- ð› ¼-L-rhamnopyranoside) (515.12_8)	0.02	0.08	0.23	0.03	0.16	0.03	0.18	0.11	0.20	0.13
Pinoresinol-glucoside (519.19_6.8)	0.30	0.05	0.10	0.09	0.03	0.13	0.03	0.82	0.25	0.17
Procyanidin B2_a (577.14_5.4)	0.04	0.00	0.00	0.00	0.00	0.01	0.00	0.00	0.00	0.00
Procyanidin B2_b (577.14_5.7)	0.00	0.00	0.00	0.00	0.00	0.00	0.00	0.00	0.00	0.00
Procyanidin B2_c (577.14_6.2)	0.01	0.00	0.00	0.00	0.00	0.00	0.00	0.00	0.00	0.00
Glucosylvitexin (593.15_6)	0.01	0.01	0.07	0.02	0.02	0.25	0.04	0.21	0.04	0.05
Kaempferol-3-O-rutinoside (593.15_5.6)	0.53	1.88	1.18	1.52	1.76	0.02	3.27	0.03	1.04	0.82
Puerarin xyloside (593.15_5.9)	0.01	0.03	0.01	0.02	0.02	0.00	0.06	0.00	0.01	0.02
Rutin (609.15_6.5)	7.07	0.00	0.00	0.11	0.00	0.00	0.00	0.01	0.01	2.35
Isorhamnetin-O-neohespeidoside (623.16_6.6)	0.63	0.00	0.00	0.00	0.01	0.00	0.00	0.00	0.00	0.09
Quercetin-di-O-glucoside (625.14_5.4)	0.03	0.02	0.02	0.03	0.02	0.04	0.02	0.04	0.02	0.02
Quercetin-di-O-glucoside 2	0.03	0.02	0.02	0.01	0.01	0.01	0.01	0.02	0.01	0.02

	**Colombia**	**Fiji**	**India**	**Kenya**	**Madagascar**	**Mexico**	**Philippines**	**Reunion**	**South Africa**	**UK**
**Phenolic acids %**										
2,4-Dihydroxybenzoic acid (153.02_6)	0.01	0.01	0.00	0.01	0.02	0.01	0.00	0.01	0.00	0.00
Protocatechuic acid (153.02_5)	0.00	0.01	0.01	0.01	0.01	0.01	0.01	0.02	0.01	0.03
Shikimic acid (173.05_2)	0.01	0.03	0.02	0.02	0.03	0.02	0.02	0.03	0.02	0.01
Quinic acid (191.06_1.9)	0.31	0.80	0.36	0.60	0.64	0.11	0.20	0.48	0.36	0.35
Ferulic_Isoferulic acid (193.05_1.9)	0.00	0.00	0.00	0.00	0.01	0.00	0.00	0.00	0.00	0.00
Chlorogenic acid 1 (353.09_5)	4.26	7.37	4.45	7.18	6.86	5.13	3.90	7.77	6.14	1.86
Cryptochlorogenic acid 1 (353.09_5.4)	1.31	1.75	0.86	1.15	0.94	0.84	0.87	2.19	1.29	2.26
Neochlorogenic acid 1 (353.09_5.3)	0.04	0.01	0.01	0.07	0.01	0.01	0.01	0.02	0.01	0.01

	**Colombia**	**Fiji**	**India**	**Kenya**	**Madagascar**	**Mexico**	**Philippines**	**Reunion**	**South Africa**	**UK**
**Others %**										
Hydroxytyrosol a (153.06_4.8)	0.03	0.05	0.02	0.04	0.02	0.02	0.02	0.08	0.03	0.07
p-Coumaric acid (163.04_5.3)	0.14	0.10	0.11	0.25	0.14	0.16	0.13	0.10	0.11	0.04
Phenprobamate (164.07_4.6)	0.06	0.60	0.12	0.13	0.08	0.13	0.32	0.69	0.37	0.54
Vitamin C (175.02_6.1)	0.04	0.06	0.05	0.12	0.04	0.05	0.04	0.04	0.04	0.00
Azelaic acid (187.1_6.9)	0.12	0.30	0.15	0.33	0.09	0.09	0.16	0.23	0.30	0.37
Citric acid (191.02_2.1)	0.01	0.00	0.00	0.00	0.00	0.00	0.00	0.00	0.00	0.02
L-Tryptophan (203.08_5.2)	0.11	0.49	0.10	0.19	0.07	0.12	0.14	0.88	0.41	0.60
Pantothenic acid (218.1_4.7)	0.03	0.04	0.01	0.02	0.05	0.05	0.11	0.06	0.02	1.64
Traumatic acid iso a (227.13_8.7)	0.10	0.54	0.12	0.27	0.08	0.06	0.29	0.71	0.37	0.15
cis- Traumatic acid iso b (227.13_8.5)	0.21	0.20	0.16	0.35	0.07	0.10	0.08	0.31	0.17	0.15
Traumatic acid iso c (227.13_8.3)	0.03	0.05	0.03	0.05	0.02	0.02	0.03	0.11	0.06	0.07
Myristic acid (227.2_12.8)	0.06	0.04	0.07	0.07	0.05	0.05	0.07	0.07	0.07	0.14
Uridine (243.06_2.1)	0.08	0.12	0.09	0.16	0.06	0.20	0.10	0.13	0.10	0.16
Alpha-Linolenic acid (277.22_12.9)	14.41	23.88	18.17	31.91	19.00	20.05	21.91	23.57	22.63	19.19
Linoleic acid (279.23_13.3)	5.32	9.65	7.31	11.55	8.06	9.49	9.45	8.89	8.40	9.14
Stearic acid (283.26_14.4)	0.53	1.25	0.47	1.31	0.89	0.71	1.34	1.08	0.96	1.54
Arachidic acid (311.3_15)	0.16	0.42	0.16	0.42	0.27	0.19	0.41	0.37	0.35	0.61
Vitamin B2 (375.13_5.6)	0.00	0.00	0.00	0.00	0.00	0.00	0.00	0.00	0.00	0.05
Oleanolic acid/Betulinic acid (455.35_12.9)	0.05	0.01	0.02	0.15	0.12	0.04	0.01	0.04	0.08	0.06

**Table 3 ijms-25-08109-t003:** Relative percentages of the glucosinolates/isothiocyanates, flavonoids, and phenolic acids in MOLE.

Mole	Glucosinolates (%)	Flavonoids (%)	Phenolic Acids (%)	Others (%)
Colombia	**50.81** ± 0.32	**21.78** ± 0.21	**5.94** ± 0.22	**21.47** ± 0.11
Fiji	**36.38** ± 0.57	**15.84** ± 0.12	**9.99** ± 0.17	**37.80** ± 0.12
India	**49.89** ± 0.01	**17.24** ± 0.03	**5.71** ± 0.04	**27.16** ± 0.31
Kenya	**24.61** ± 0.03	**19.01** ± 0.01	**9.05** ± 0.02	**47.33** ± 0.03
Madagascar	**32.31** ± 0.04	**30.09** ± 0.01	**8.53** ± 0.03	**29.08** ± 0.09
Mexico	**48.19** ± 0.03	**14.18** ± 0.12	**6.13** ± 0.02	**31.52** ± 0.05
Philippines	**42.42** ± 0.03	**17.97** ± 0.01	**5.01** ± 0.01	**34.60** ± 0.06
Réunion	**34.99** ± 0.06	**17.13** ± 0.01	**10.51** ± 0.01	**37.37** ± 0.05
South Africa	**35.16** ± 0.06	**22.57** ± 0.01	**7.84** ± 0.01	**34.48** ± 0.05
UK	**42.11** ± 0.07	**18.82** ± 0.01	**4.53** ± 0.01	**34.55** ± 0.07

## Data Availability

Data are contained within the article.

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
