# Peer review of "Comparative Metabolomic Analysis of Moringa oleifera Leaves of Different Geographical Origins and Their Antioxidant Effects on C2C12 Myotubes"

_ijms, 2024, doi:10.3390/ijms25158109_

Round 1

Reviewer 1 Report

Comments and Suggestions for Authors

In the manuscript “Comparative metabolomic analysis of Moringa oleifera leaves from different geographical origins and their antioxidant effects on C2C12 myotubes” Ceci et al. present an antioxidant capacity and activity analysis of Moringa oleifera from 4 continents and 9 countries. The introduction and discussion sections of this paper are well-written and provide a comprehensive background and context for the research, highlighting its significance and potential impact. However, to enhance the robustness and significance of the in vitro part of the paper, it is recommended that additional experiments are conducted. These further investigations will not only validate the initial findings but also provide a more comprehensive understanding of the research topic.

Comments on the Quality of English Language

The results section would benefit from English editing to improve the clarity of the presented findings. 

Reviewer 2 Report

Comments and Suggestions for Authors

file attached

Reviewer 3 Report

Comments and Suggestions for Authors

The study aims to differentiate samples of Moringa oleifera from various geographical origins. This is an interesting approach to evaluating the diversity of this plant. The work is well-designed, and the results are well discussed.

Minor Comments:

  1. In Section 2, please specify the method used to determine antioxidant activity.
  2. Renumber the figures after Figure 7.
  3. On page 10, lines 343 and 349 – please replace "Figure 6" with "Figure 8."
  4. The figure referred to as Figure 7 (correctly Figure 9) is not mentioned in the text.
  5. In Section 4.4 on page 16, clarify that the antioxidant activity is expressed in micromoles per milligram of protein (µmol/mg protein).
Comments on the Quality of English Language

Minor editing of English language required

Round 2

Reviewer 1 Report

Comments and Suggestions for Authors

In the manuscript Ceci et al. present an antioxidant capacity and activity analysis of Moringa oleifera from 4 continents and 9 countries. As mentioned previously the introduction and discussion sections of this paper are well-written and were further improved by the editing and expansions performed. Unfortunately, the absence of a point-by-point response to the comments in the first review makes it more difficult to assess the extent of the changes made in the updated version.

Attached are my remaining concerns with the manuscript.

Comments on the Quality of English Language

Author Response

Reviewer 1

In the manuscript “Comparative metabolomic analysis of Moringa oleifera leaves from different geographical origins and their antioxidant effects on C2C12 myotubes” Ceci et al. present an antioxidant capacity and activity analysis of Moringa oleifera from 4 continents and 9 countries. As mentioned previously the introduction and discussion sections of this paper are well-written and were further improved by the editing and expansions performed.

We thank the Reviewer for his positive appreciations and comments that help us improve the quality of our manuscript.

All changes made in the revised version of the manuscript are highlighted in red.

Unfortunately, the absence of a point-by-point response to the comments in the first review makes it more difficult to assess the extent of the changes made in the updated version.

Below are my remaining concerns with the manuscript.

We apologize for any inconvenience.

A point-by-point response regarding the first revision was included.

In any case, we thank the reviewer for the note and now we have modified the manuscript in accordance with his kind suggestions.

Major concerns:

1 - You performed thorough analysis of Moringa oleifera from different locations, the paper would benefit from images and/or description of plants characteristics after being grown in same location and whether differences were observed (for example in leaf specimens that were used for extract).

We thank the Reviewer for highlighting this important issue, which requires, ideally, coordinated effort by the global moringa community to address fully. We added in the appropriate place the text "Our samples describe much of the variation that is well-documented within M. oleifera, e.g. spanning variants with short fruits with few, large seeds to those with long fruits and many small seeds; leaves with greater or lesser degrees of red pigment on the rachis; or variation in flower color, from white to cream (e.g. Abubakar et al. 2011, Mobarak et al. 2017, Popoola et al. 2016). The individuals we studied here have been previously examined for variables such as protein content and glucosinolate profiles, and they do show some variation (Chodur et al. 2018), though accessions of M. oleifera are much more similar to one another than they are to other Moringa species (e.g. Olson et al. 2016, Fahey et al. 2018)" (line 590).

Abubakar, B.Y., MuA’zu, S., Khan, A.U. and Adamu, A.K., 2011. Morpho-anatomical variation in some accessions of Moringa oleifera Lam. from Northern Nigeria. African Journal of Plant Science5(12), pp.742-748.

Mobarak, A.A., Shaltout, K., Ali, H.I., Baraka, D. and Aly, S., 2017. Morphological VariabilityAmong Moringa oleifera (Lam.) Populations in Egypt. Egyptian Journal of Botany57(1), pp.241-257.

Popoola, J.O., Bello, O.A. and Obembe, O.O., 2016. Phenotypic intraspecific variability among some accessions of drumstick (Moringa oleifera Lam.). Canadian Journal of Pure and Applied Sciences10(1), pp.3681-3693.

2- The paper would benefit from inclusion of a clear schematic representation of the workflow detailed in the manuscript.

We thank the Reviewer for the suggestion. We have included a schematic representation of the workflow as figure 10.

3- Please add details about the number of specimens analyzed for data in Table 2 in figure legend, as well as SD for each metabolite.

We thank the Reviewer for the suggestion. We have now inserted the table of all metabolites as Table 2 and modified Table 3 (ex-Table 2) as kindly suggested.

4- The addition of MTS analysis and microscopic images strengthens the viability results presented in the paper, and the manuscript would benefit from inclusion of this analysis either as part of the main text or in the supplementary materials for readers to access.

We thank the Reviewer for the suggestion. We have included the MTS test and microscope images as supplementary material 1.

5- How might the growth of Moringa oleifera in its original regions, as opposed to the same environmental conditions tested in the paper, could affect the results? Please address this point in the discussion section.

This is a very important question. First, Moringa oleifera is not known in the wild, so its truly original regions are unkown, though they almost certainly fall somewhere in what is today India. Second, the traditional area of cultivation, as we allude to in the manuscript, has expanded slowly across the earth over probably thousands of years. Moringa reached the Pacific coast of Mexico, where our study site was located, 300 years ago at the latest, and therefore unambiguously falls within the traditional area of spontaneous cultivation. Because there is no ancestral highly localized point of reference, we suggest that the salient question here is not "our site vs an ancestral point" but instead "how much of the variation detected in traits of applied interest across M. oleifera studies is due to local adaptation; how much is due to cultivation conditions, including soil; and how much is due to processing practices (this last consideration is probably even more important than the former)?" See the exhortation to such global studies in Olson (2017).

In the light of the reviewer's comment, we have gone over the manuscript to make sure that it is clear that there are many moringa studies in the literature and that they show a fair degree of variation across them. These all involve local germplasm, e.g. a study of moringa in northern Nigeria; a study of moringa in a small part of Nicaragua; a study of moringa in Egypt, etc. In this context, our common garden study is designed to highlight this major gap in the literature, which is the difficulty of interpreting results from different regions using different methods and processing techniques. Our study using not only a common garden but identical processing sheds light on this issue. Please see the comment we have added along these lines on line 546.

Olson, M. E. 2017. Moringa frequently asked questions. Acta Horticulturae 1158 1158: 19-32.

Minor concerns:

1- Please rephrase the introduction paragraph between lines 89-103 to exclude the expression “extreme”.

We thank the reviewer for the suggestion. We changed the text accordingly (line 89-92 and 103-105).

2- Please add referral to supplementary material attached in the results section of the paper.

We thank the Reviewer for the note. We changed the text accordingly (line 368).

3- Add ref to statement in line 138

We thank the Reviewer for the note. We have inserted a new reference for this statement (line138).

4- Corrects figure number in line 352.

We apologize for the typos/editorial errors in the text. We changed the text accordingly (line 356).

5- In section 2.3 rephrase “various dilutions of MOLE stock solution” to “dilution of various MOLE stock solution”

We thank the Reviewer for the note. We changed the text accordingly (line 357).

6- Rephrase line 417-420 to increase clarity

We thank the Reviewer for the note. We changed the text accordingly (line 419-424).

7- Rephrase line 641 to increase clarity

We thank the Reviewer for the note. We changed the text accordingly (line 664).

Reviewer 2 Report

Comments and Suggestions for Authors

file attached

Author Response

Reviewer 2

We thank the Reviewer for his positive appreciations and comments that help us improve the quality of our manuscript.

All changes made in the revised version of the manuscript are highlighted in red.

“The table should be included in the results section to clearly display the chemical composition of each tested plant. This will facilitate the analysis of the correlation between the biological activity and the chemical composition. We observe that the results of all nine MOLE samples are very similar, indicating that the chemical composition must be responsible for these consistent outcomes.”

We thank the Reviewer for the note. We have now inserted the table as kindly suggested in the manuscript as Table 2.

Round 3

Reviewer 2 Report

Comments and Suggestions for Authors

My request was:

The table should be prepared (not to insert the raw data from the instrument) and included in the results section to display the chemical composition, percentage, and match factor of the tested plants.

You should then discuss the correlation between the chemical compounds and the results of the biological tests.

and none of this was done

Author Response

Reviewer 2

We thank the Reviewer for his positive appreciations and comments that help us improve the quality of our manuscript.

All changes made in the revised version of the manuscript are highlighted in red.

“The table should be prepared (not to insert the raw data from the instrument) and included in the results section to display the chemical composition, percentage, and match factor of the tested plants.”

According to the previous reviewer’s suggestion we have inserted the requested table with the list of 77 identified metabolites in the main text of the manuscript as table 2 in the results section.

We apologize for our mistake in the first version of the table. Unfortunately, we missed to insert the specification about how the values are expressed. Table 2 represents the percentage of each individual metabolite compared to the totality of the peaks detected and was produced after the elaboration of the data peaks detected by the instrument. It is therefore not a direct expression of the raw data but an elaboration on percentage values.

Each column represents a different sample (in alphabetical order from left to right), and the relative percentage of the metabolites present in it.

For clarity of information, we have now amended the table.

We have modified the heading and reduced the number of decimals present in the numerical values ​​in order to make the percentage distribution clearer and the most represented metabolites more evident.

The metabolites were divided by groups of molecules (glucosinolates/isothiocyanates, flavonoids, phenolic acids and other compounds). The total percentage of molecules present in the different groups is shown in table 3.

“You should then discuss the correlation between the chemical compounds and the results of the biological tests.”

We thank the Reviewer for the note.

Metabolomics analysis of our samples highlighted a similar profile of the identified metabolites.

However, differences between the different extracts were found when considering the relative percentages of the different metabolite groups (glucosinolates/isothiocyanates, flavonoids, and phenolic acids) in MOLE samples (table 3).

For this reason, we have chosen to comment, in the discussion section, the effects of metabolites for groups of molecules (lines 461-517) with particular reference to some specific literature evidence.

We consider that the biological effect of the extracts is due to a synergistic effect of the mixture of all the bioactive molecules altogether.

In this work we did not test the action of a single metabolite but the effect of the mixture.

We thank the reviewer for the comment. We are thinking to start a couple of projects focused on the administration of some purified metabolites by evaluating their antioxidant action in cultured muscle cells.

For clarity, a comment has been added to the discussion section (line 546-559).

Round 4

Reviewer 2 Report

Comments and Suggestions for Authors

Accept in present form